# The influence of survival conditions on farmer willingness to participate in non-grain conversions of cultivated land based on the SOR model

Hui Fan [1]*, Jiaxin Liu[2], Xiaoke Shao[1], Bingqian Sun[2], Yilin Yang[1], Chaoge Shi[1]

**1** College of Geography Science, Xinyang Normal University, Xinyang, Henan, China, **2** Key Research Institute of the Yellow River Civilization and Sustainable Development, Henan University, Kaifeng, Henan, China

* fanhuie2002@163.com

## Abstract

Non-grain conversions (NGC) of cultivated land pose a great threat to national food security, and farmers are the direct actors in the NGC of cultivated land. Exploring the influence of farmer' survival situation on their willingness to convert cultivated land to non-grain land can provide theoretical support for formulating control policies for NGC of cultivated land. Based on the data obtained from 650 questionnaires in Henan Province and using the SOR model as the theoretical basis, this study explored the impact mechanism of the survival situations of two types of farmers, namely, those with agricultural livelihoods (AL) and those with nonagricultural livelihoods (NL), on their willingness to convert farmland into non-grain land through the use of structural equation models. The results are as follows. The perceived benefits (PB) of NGC can improve their willingness to convert farmland to non-grain land, while the perceived risks (PR) of NGC reduce their willingness to convert farmland to non-grain land. The influence of PB and PR on farmer willingness to engage in NGC of cultivated land in the near term is greater than that in the long term. The influence mechanism of perceived income and perceived risk on the willingness of non-grain conversions of cultivated land varies with the livelihood types of the farmers. The influences of internal family situation (IS) and external family situations (ES) on the willingness to engage in NGC of cultivated land differ according to the types of farmer livelihood. There are differences in the influence mechanism of survival situations on different types of farmers' willingness to non-grain conversion of cultivated land.

## Introduction

Food forms the foundation of national prosperity and is one of the key resources for the continuation of civilization. Food security is "the greatest concern of a nation" [1]. Governments around the world have implemented food security as an important strategy. In recent years, China's urbanization and industrialization have developed rapidly, and remarkable achievements have been made in economic construction. However, these achievements have

**Data availability statement:** All the data related to this study have been included in the paper and its Supporting Information files.

**Funding:** The National Natural Science Foundation of China (Grant number 71904150, 41771438) and General Project of Humanities and Social Sciences Research in Higher Education Institutions in Henan Province (Grant number 2024-ZZJH-039).

**Competing interests:** The authors have declared that no competing interests exist.

**Abbreviation:** NGC, Non-Grain Conversions; SOR, Stimulus-Organism-Response; SEM, Structural Equation Models; WNG, willingness to non-grain conversions of cultivated land; AL, Agricultural Livelihoods; NL, Nonagricultural Livelihoods; PB, Perceived Benefits; PR, Perceived Risks; IS, Internal Family Situation; ES, External Family Situations; RW, Recent Willingness; LW, Long-term Willingness.

also brought great challenges to cultivated land protection and food security. The continuous spread of cities has threatened regional food security [2]. The rapid decrease in the scale of cultivated land in China has increased the risks to resources and food security [3]. From a strategic perspective, the Chinese government has the responsibility to ensure the country's food security and to make food production self-sufficient [4]. The Chinese government has always attached great importance to food security. The Central Rural Work Conference held at the end of 2022 and the No. 1 Central Document of 2023 contain important instructions on food security and cultivated land protection. Currently, food security is facing many challenges, especially the NGC of cultivated land. Research shows that the NGC rate of cultivated land in China is approximately 27%, among which the proportion of NGC of cultivated land in Southwest China is as high as 46% [5].

In recent years, academics have explored non-grain cultivated land in depth and achieved fruitful results. The existing research results mainly include three parts, namely, the spatial distributions, the driving mechanisms and the effects for NGC. First, the spatial distributions of NGC were explored. The NGC rate of cultivated land in western China is relatively high, while it is relatively low in the eastern part of China, with significant spatial differences and aggregation distributions [6]. Sun et al. [7] believe that the lower economic benefits of the planting industry are the most important driving force, leading to spatial differences in NGC of cultivated land. The spatial characteristics of different types of NGC of cultivated land [8]. A new measurement method was used to explore the spatial pattern of NGC of cultivated land in Hubei Province over the past 30 years [9].Second, the driving mechanism of NGC. Economic interests are the most fundamental mechanism to induce NGC on cultivated land [10]. The expansion of areas with NGC in different periods in China is closely related to the attitudes of the central government and local governments [11]. The transformation of NGC of cultivated land has multiple driving forces [12]. Zhang et al. [13] noted that different industrial policies determined the NGC mode at different stages. The driving factors for NGC of cultivated land vary among different land leasing entities [14]. The driving mechanisms for NGC of different types of cultivated land [15]. The driving mechanism of NGC of cultivated land between compensated cultivated land and original cultivated land in mountainous areas [16]. Third, the effects of NGC of cultivated land. Lu et al. [17] explored the impact of NGC of cultivated land on the potential for food production. The NGC of cultivated land has an impact on agricultural carbon supply and demand [18], as well as carbon emission intensity [19]. The NGC of cultivated land is closely related to agricultural subsidies [20], and so does the level of intensive utilization of cultivated land [21]. In addition, The other issues of NGC of cultivated land has also attracted the attention of the academic community. Chen Meiqiu [22] suggested that different types of NGC should be treated and handled differently. The implementation of basic farmland and high-standard basic farmland policies did not stop the NGC of cultivated land but actually promoted NGC [11].

The existing research results provide a basis for the in-depth exploration of NGC of cultivated land, but there are also some shortcomings. First, the exploration of NGC of cultivated land—farmers—has been ignored. Farmers are the direct actors in the NGC of cultivated land, and they decide whether to convert farmland into non-grain land on the basis of weighing their livelihood security and economic efficiency [23]. Second, there is a lack of cooperative exploration of the macro- and micro conditions of NGC on cultivated land. NGC of cultivated land is a type of land use behavior for farmers under the joint action of macro-conditions (ES) and micro-conditions (IS). ES and IS have an impact on farmers' willingness to NGC of cultivated land. Third, the analysis of farmer heterogeneity is insufficient. Because of the large size of farmer groups and their different educational levels, family livelihoods, family assets, cognitive levels and other factors, it is necessary to explore the influence mechanism of

NGC of cultivated land from the perspective of farmer heterogeneity. The individual characteristics of farmers affect their willingness to NGC of cultivated land. There are differences in the behavior mechanisms of NGC of cultivated land among farmers of different ages and working conditions [24]. Family status has an impact on farmers' NGC of cultivated land. The household life cycle of farmers affects the restraining effect of agricultural subsidies on NGC of cultivated land [20]. Household labor and capital also affect farmers' willingness to NGC of cultivated land [25]. In this study, the scientific problem is the influence mechanism of survival situations on farmers' willingness to NGC of cultivated land. Although the types of farmers' livelihood can be divided from different perspectives, Chinese academic circles usually divide farmers' livelihood types into agricultural livelihood and non-agricultural livelihood (Xiao Yang et al., 2024). Based on the above achievements and the actual situation of rural areas in Henan province, the study divides farmers into agricultural livelihood farmers and non-agricultural livelihood farmers from the perspective of livelihood types. Henan Province is the main grain production area in China, and its grain output has a significant impact on the national food security. The proportion of farmers engaged in agricultural production is high in Henan Province. At present, the phenomenon of NGC of cultivated land is very serious in Henan Province. In 2020, the proportion of NGC of cultivated land is about 30% in the province [26,27]. It poses a major threat to the national food security. Using Henan Province as the research area, based on a questionnaire survey of 650 sample farmers using the SOR model as the analytical framework and adopting the structural equation model (SEM), this study explores the influence mechanism of the ES and IS on farmer willingness levels to convert farmland into non-grain land to provide theoretical support for local governments to formulate control policies for the NGC of cultivated land and to mobilize farmer enthusiasm for grain production. This study aims to explore the uniqueness of the mechanism influencing the willingness to NGC of cultivated land through an in-depth examination of a specific province, Henan Province, one of the main grain producing areas, to provide theoretical support for the control of NGC of cultivated land in other similar areas.

## Theoretical analysis and research hypothesis

### Theoretical basis

The Stimulus-Organ-Response (SOR) model provides a theoretical framework for understanding how environmental stimuli (S) affect the internal state of an organism (O), leading to a behavioral response (R). This model holds that the response of individuals after receiving external stimuli is not mechanical and passive, and people have subjective initiative, and have the ability to process effective information to make rational behavior decisions under stimulation. This model is widely used in Marketing, education, health behavior, environmental psychology, For example, team work satisfaction under supervision [28], the relationships between consumer confidence and green purchase intentions [29], the influencing factors of the continuous purchase of fashion products [30], impulse purchases in online broadcasts [31], impulse purchases in omnichannel retail [32], Chinese individual outbound travel [33], and the impact of public climate policy on low-carbon travel [34].

SOR model is mainly used to explain the action willingness of the actor, and the above research proves that it has the strong explanatory power. The research team found in the interviews conducted in the early stage of the study that farmers' willingness to NGC of cultivated land followed three stages: external stimulation, information processing and reaction. Therefore, SOR model can be used to explain the mechanism of farmers' willingness to non-grain conversions.

In this study, SOR model was used to explore the factors affecting farmers' willingness to NGC of cultivated land. Under the background of NGC willingness of cultivated land,

the policy, market and the internal situation of peasant families act as the stimulus, and the perception of farmers' own income and the perception of external economic risk act as the organism, and the decision to NGC of cultivated land is the reaction. Through the application of SOR model, This can deeply understand how the policy incentives affect the farmers' mental process and perception, and then affect their decisions of NGC of cultuvated land. This will help formulate more effective policy measures to promote the rational use of land resources and the sustainable development of agriculture.

In this study, IS and ES affecting farmers' willingness to NGC of cultivated land are discussed in depth. Specifically, at the macro level, this study selected current policy, local government, benefit from cultivation and tillage conditions as the key indicators to reflect ES of farmers. At the micro level, family expenditure, family livelihood, children's expenses and medical care for the elderly ware selected as the key indicators to reflect IS of farmers. In terms of PB, economic benefits, quality of life, children's development and other dimensions were selected. In terms of PR, legal punishment risk, natural disaster risk, technical shortage risk, and technical shortage risk ware selected to reflect.

## Research hypothesis

The formation of farmer willingness to convert farmland to non-grain land goes through three stages, namely, "stimulus", "organism" and "reaction", which are the material stimulation of ES and IS, farmers' own PB-PR assessments and willingness to NGC (Fig 1). Therefore, the SOR model can be used to analyze problems such as the farmer willingness to NGC and provides a new research perspective for discussing the farmer willingness to NGC.

**The survival situations of farmers and PB-PR of NGC.** First, IS and ES of farmers affect PB and PR of NGC. Macro policies can be understood and transformed into a series of actions by farmers at the micro level [35]. When farmers evaluate PB and PR of NGC, they not only consider the survival situations that exist inside the family but also include the macro survival situations outside the family in their evaluation system.

Second, farmer perceptions of PB and PR of NGC are based on their understanding of ES and IS of their families and are not simple conditional responses. When farmers evaluate the PB-PR of NGC of cultivated land, they do not ignore they are not independent of the survival situations inside and outside the family but make PB-PR assessments according to the survival situation of the family. Based on the above theoretical analysis, the following hypotheses are proposed in this study:

**H1**: *IS has an impact on PB of NGC by farmers.*

**H2**: *ES has an impact on PB of NGC by farmers.*

**H3**: *IS has an impact on PR of NGC by farmers.*

**H4**: *ES has an impact on PR of NGC by farmer.*

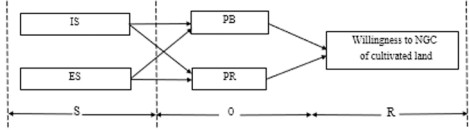

**Fig 1. Theoretical analysis framework.**

**PB-PR and willingness to NGC of cultivated land by farmers.** The willingness to NGC of cultivated land of farmers comes from the comprehensive measurement of farmers' PB and PB of NGC. In essence, NGC of cultivated land is an economic behavior. The economic interests is the most fundamental inducing mechanism for NGC of cultivated land [10]. Farmers' willingness to NGC of cultivated land is a decision after comprehending PB and PR of NGC of cultivated land. Therefore, the following hypotheses are proposed:

> **H5**: *The PB of NGC of cultivated land have an impact on farmers' willingness to NGC of cultivated land.*

> **H6**: *The PR of NGC of cultivated land have an impact on farmer' willingness to NGC of cultivated land*

**The type of household livelihood and the willingness to NGC of cultivated land.** Household livelihoods are the ways in which individuals or families make a living, and they depend on the assets of the individual or family, which the activities they engage in, and the ability to access these assets and activities [36]. The type of family livelihood not only reflects the survival situations of the farmers but also influences the understanding of the farmers' ES. At the same time, the PB-PR assessments of farmers on NGC is greatly affected by the types of family livelihood, which leads to differences in NGC of cultivated land. Therefore, the following hypothesis is proposed:

> **H7**: *Farmers with different livelihood types have different evaluations of IS and ES and different perceptions of the PB-PR of NGC of cultivated land, which lead to different intentions for NGC of cultivated land.*

In conclusion, farmers' willingness to NGC of cultivated land is influenced by many factors. Based on this, this paper constructs the theoretical framework as follows (Fig 2).

## Data sources, selection of variables and research methods

**Data sources.** Henan Province is an important grain-producing area in China. The total grain output of the province was 135.787 billion jin, ranking second in the country in 2022. The problem of NGC of cultivated land in Henan Province will have a serious impact on the national food security. The research group conducted the questionnaire survey in January and February 2024 in southern Henan (Huaibin County, Xinyang city), eastern Henan (Xihua County, Zhoukou city), northern Henan (Linzhou city, Anyang city), western Henan (Yiyang County, Luoyang city) and central Henan (Dengfeng city, Zhengzhou city) (Fig 3). The combination of stratified sampling and random sampling was used to determine a variety of target sample numbers in a county according to the various registered agricultural populations in the county. Then, 3 to 4 townships in each sample county were randomly selected by the numbering method, and 3 to 5 villages in each sample

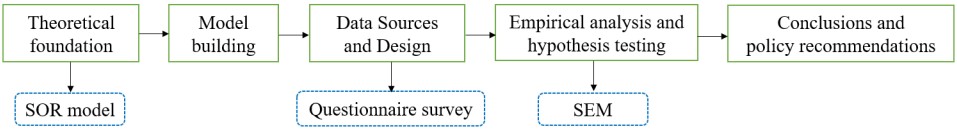

**Fig 2. Technical flow chart.**

township were also randomly selected for the investigation. The total of 680 questionnaires was distributed in this study, and 650 valid questionnaires were obtained, for the effective response rate of 95.6%. In order to ensure the accuracy of the questionnaire data, a one-on-one survey was used in the questionnaire survey, and questionnaires with missing observation values, options that are not in line with the reality of farmers and other potential problems were eliminated.

According to the statistical analysis of the sample farmers, there were no significant differences in gender, age, and distance from the village to township government, or farmland plots per household between the agricultural families and nonagricultural families (Table 1). However, the two types of sample farmers have certain differences in the average cultivated land area per household. The average cultivated land area per household of AL sample farmers (6.1 mu/household) is larger than that of NL sample farmers (4.0 mu/household). The difference

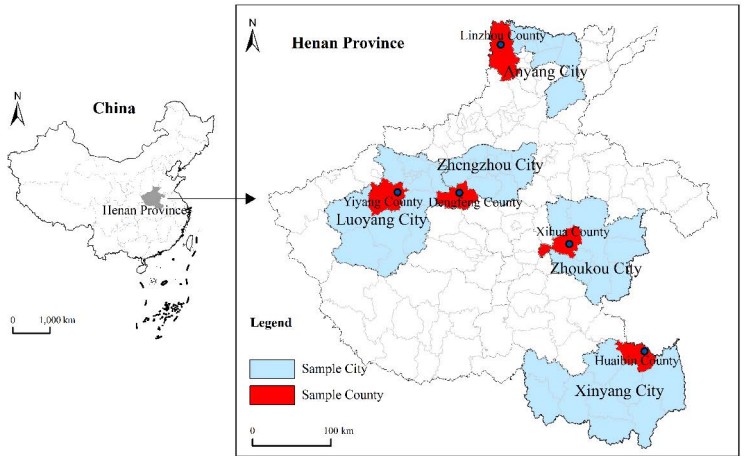

**Fig 3. Overview map of the study area.**

**Table 1. Statistical analysis of sample farmers.**

| Index | Identity | AL | | NL | |
|---|---|---|---|---|---|
| | | Frequency/household | Weight/% | Frequency/household | Weight/% |
| Gender | Man | 135 | 44.41 | 153 | 44.22 |
| | Woman | 169 | 55.59 | 193 | 55.78 |
| Age/year | ≤18 | 9 | 2.96 | 56 | 16.18 |
| | 19 ~ 40 | 74 | 24.34 | 158 | 45.66 |
| | 41 ~ 65 | 154 | 50.66 | 106 | 30.64 |
| | ≥66 | 67 | 22.04 | 26 | 7.51 |
| Distance from village to township government/km | ≤3 | 100 | 32.89 | 107 | 30.92 |
| | 3 ~ 6 | 123 | 40.46 | 139 | 40.17 |
| | ≥6 | 81 | 26.64 | 100 | 28.91 |
| Area of cultivated land per household | Acres/household | 6.1 | | 4.0 | |
| Number of cultivated land pieces per household | Block/household | 3.0 | | 3.3 | |
| Recent willingness (RW) of NGC | Yes | 136 | 44.74 | 121 | 34.97 |
| | No | 168 | 55.26 | 225 | 65.03 |
| Long-term willingness (LW) of NGC | Yes | 129 | 42.43 | 166 | 47.98 |
| | No | 175 | 57.57 | 180 | 52.02 |

between the two types of sample farmers is significant in terms of RW and LW for NGC of cultivated land. According to the longitudinal comparison, RW of NGC for the AL sample farmers is high, while LW of NGC for the NL sample farmers is relatively low. The situation for NL sample farmers is exactly the opposite; that is, RW of NGC for the NL sample farmers is low and LW of NGC is high for the NL sample farmers. According to the horizontal analysis, RW of NGC for the AL sample farmers is greater than those of the NL sample farmers. However, LW of NGC for the AL sample farmers is lower than those of the NL sample farmers.

## Variable selection and assignment

According to the above analysis and combined with the research objectives, the initial measurement items are formulated. Combined with the results of the pre-survey and interviews and based on the opinions of relevant experts, the individual questions in the questionnaire were optimized to form the final scale (Table 2). The questionnaire uses a five-point Likert scale.

ES and IS have a greater impact on farmers with AL, while the impact on farmers of NL is relatively small (Table 2). According to the statistical index of PB, the income evaluation of NGC of cultivated land by AL farmers is slightly higher than that of NL farmers. The AL farmers believe that the PR of NGC of cultivated land is greater. However, the NL farmers think that the PR of NGC of cultivated land is small.

## Research methods

In social science research, some concepts (i.e., latent variables) are difficult to measure directly and accurately, but some explicit indicators can be used to measure these latent variables indirectly. Previous statistical methods cannot meet the demand of measuring latent variables, but structural equation models (SEM) can measure the latent variables and their related indicators together. SEM consists of a measurement equation and a structural equation. The measurement model is expressed as follows:

$$x = \Lambda_x \xi + \delta \tag{1}$$

$$y = \Lambda_y \eta + \varepsilon \tag{2}$$

The structural equation expression is as follows:

$$\eta = B\eta + \Gamma \xi + \zeta \tag{3}$$

where $x$ is the exogenous observed variable, $\xi$ is the vector of the exogenous latent variable, and $\Lambda_x$ is the factor load matrix of the exogenous latent variables and exogenous observed variables. Delta is the residual matrix in Equation (1), $y$ is the endogenous observed variable, $\eta$ is the vector of endogenous latent variables, and $\Lambda_y$ is the factor load matrix of the observed escape quantity of the endogenous latent variable. $\varepsilon$ is the error term of Equation (2), $B$ is the coefficient matrix of the endogenous latent variables, and $\Gamma$ is the effect of the endogenous latent variable on the endogenous latent variable. $\zeta$ is the error term of the structural equation.

## Results and analysis

### Model fitting

**Reliability test.** This test mainly reflects the internal consistency of the latent variables (Cronbach's α). Using the reliability analysis function of SPSS 20.0, the consistency coefficients

**Table 2. Measurement variables and descriptive statistics.**

| Latent Variables | Observation Variables | Observation Variables meaning and value of question items | AL | | NL | |
|---|---|---|---|---|---|---|
| | | | Mean value | Standard deviation | Mean value | Standard deviation |
| ES | Current policy/ ES1 | Current policy's impact on NGC of cultivated land: very serious = 1;relatively serious = 2;general = 3;relatively slight = 4;very slight = 5 | 2.52 | 0.87 | 2.72 | 0.83 |
| | Local government/ ES2 | Local government's impact on NGC of cultivated land: same as above. | 2.75 | 0.86 | 2.86 | 0.82 |
| | Benefit from cultivation/ES3 | Benefit from cultivation on NGC of cultivated land: same as above. | 2.41 | 1.09 | 2.62 | 1.01 |
| | Tillage conditions/ ES4 | Tillage conditions on NGC of cultivated land: same as above. | 2.65 | 0.78 | 2.77 | 0.76 |
| IS | Family expenditure/IS1 | Family expenditure on NGC of cultivated land: same as above. | 2.53 | 0.94 | 2.81 | 0.84 |
| | Family livelihood/ IS2 | Family livelihood on NGC of cultivated land: same as above. | 2.54 | 0.95 | 2.76 | 0.94 |
| | Children's expenses/IS3 | Children's expenses on NGC of cultivated land: same as above. | 2.84 | 0.89 | 3.06 | 0.84 |
| | Medical care for the elderly/IS4 | Medical care for the elderly on NGC of cultivated land: same as above. | 2.73 | 0.86 | 3.06 | 0.76 |
| PB | Increase economic benefits/PB1 | NGC of cultivated land is beneficial to increase economic benefits: very agree = 1; comparative identity = 2;general = 3; comparative disapproval = 4;very disagree = 5 | 3.59 | 0.95 | 3.23 | 0.79 |
| | Improve quality of life/PB2 | NGC of cultivated land is conducive to improving quality of life: same as above. | 3.47 | 0.85 | 3.15 | 0.74 |
| | Promote children's development/PB3 | NGC of cultivated land is conducive to promoting children's development: same as above. | 3.11 | 0.75 | 2.80 | 0.70 |
| | Improve social status/PB4 | NGC of cultivated land is conducive to improving social status: same as above. | 3.05 | 0.78 | 2.74 | 0.78 |
| PR | Legal unishment risk/PR1 | NGC of cultivated land is faced with the risk of legal punishment: same as above. | 2.51 | 0.84 | 2.69 | 0.73 |
| | Natural disaster risk/PR2 | NGC of cultivated land is faced with the risk of natural disaster: same as above. | 3.52 | 0.70 | 3.73 | 0.78 |
| | Technical hortage risk/PR3 | NGC of cultivated land is faced with the risk of technical shortage: same as above. | 3.06 | 0.76 | 3.32 | 0.73 |
| | Funds reakdown risk/PR4 | NGC of cultivated land is faced with the risk of funds breakdown: same as above. | 3.11 | 0.73 | 3.35 | 0.75 |

of the four latent variables of the two livelihood types were all greater than the threshold condition of 0.6 (Table 3). Therefore, the reliability test passes.

**Validity test.** The KMO values of each potential variable for the two livelihood types ranged from 0.618 to 0.787 (Table 3), which were greater than the threshold requirement of 0.5 and passed the Bartlett sphere test. This indicates that the model data have high validity.

**Model fitting and fit test.** AMOS 23.0 software was used to test the influencing factors for farmers' willingness to NGC of cultivated land. In SEM of the AL farmers, two sets of covariant relationships were added, namely, e1 and e2, e7 and e8, considering the existence of covariant relationships between the observed variables for ES and the observed residuals for IS, which effectively reduced the chi-square value of the model and did not deviate from the theoretical assumptions. Similarly, in SEM for the NL farmers, a set of covariant relations is added, e7 and e8, which effectively reduces the chi-square value of the model and does not violate the theoretical hypothesis. Finally, the fit indices (Table 4), model path diagrams (Figs 4 and 5) and path coefficients (Table 5) of the two types of farmer SEMs were obtained. Through a comparison with all the criteria, it is found that the fit index of the model meets the

**Table 3. Analysis of reliability and validity.**

| Latent Variable | Observation Variable | AL | | | NL | | |
|---|---|---|---|---|---|---|---|
| | | Factor Loading | *KMO* | Cronbach's α | Factor Loading | *KMO* | Cronbach's α |
| ES | ES1 | 0.748 | 0.709 | 0.789 | 0.741 | 0.710 | 0.796 |
| | ES2 | 0.694 | | | 0.742 | | |
| | ES3 | 0.757 | | | 0.775 | | |
| IS | IS1 | 0.785 | 0.787 | 0.858 | 0.793 | 0.743 | 0.865 |
| | IS2 | 0.806 | | | 0.851 | | |
| | IS3 | 0.718 | | | 0.773 | | |
| | IS4 | 0.714 | | | 0.698 | | |
| PB | PB1 | 0.756 | 0.686 | 0.816 | 0.778 | 0.663 | 0.771 |
| | PB2 | 0.823 | | | 0.874 | | |
| | PB3 | 0.815 | | | 0.801 | | |
| PR | PR1 | 0.640 | 0.618 | 0.650 | 0.702 | 0.631 | 0.641 |
| | PR2 | 0.624 | | | 0.732 | | |
| | PR3 | 0.828 | | | — | | |
| | PR4 | — | | | 0.702 | | |

**Table 4. Results of the overall model fit test.**

| Fit Index | Absolute fit indices | | | | Value added fit indices | | Simplified fit indices | |
|---|---|---|---|---|---|---|---|---|
| | χ2/df | GFI | AGFI | RMSEA | NFI | RFI | PGFI | PNFI |
| Evaluation criterion | <3 | >0.8 | >0.8 | <0.08 | >0.8 | >0.8 | >0.5 | >0.5 |
| AL | 2.752 | 0.904 | 0.857 | 0.079 | 0.895 | 0.864 | 0.61 | 0.69 |
| NL | 2.809 | 0.913 | 0.873 | 0.076 | 0.877 | 0.843 | 0.624 | 0.685 |

fit evaluation, and the overall fit is good. The constructed structural equation model is suitable for path analyses of the impact of survival situations on farmer willingness to NGC.

## Results analysis

**Model verification results.** The path diagram (Figs 4 and 5) and path coefficient table (Table 5) were drawn based on the fitting results of SEMs of the farmers with two types of livelihoods by using AMOS 23.0 software. The results show that the standardized path coefficients (B) of each path that influence the willingness to NGC of arable land for farmers with two types of livelihood pass the significance (P) test. The specific values of the path coefficients, B and P, for AL and NL are analyzed as follows:

The coefficients and significances of IS→PB path (the B values of AL and NL were 0.581 and −0.616, respectively; P = 0.01 and 0.01, respectively) meet the requirements, and H1 is confirmed. The coefficients and significances of ES→PB path (B: −0.729, 0.414; P: 0.01, 0.01, respectively) confirmed H2. IS→PR path coefficient and significance (B: −0.625, 0.405; P: 0.01, 0.01, respectively) confirmed Hypothesis H3. The coefficients and significances of ES→PR path (B: 0.644, −0.581; P: 0.01, 0.01, respectively) formalizes H4. The path coefficient and significance of PB→willingness to non-grain conversions of cultivated land (WNG) (B: 0.573, 0.332; P: 0.01, 0.01, respectively) confirmed H5. The path coefficient and significance of PR→WNG (B: −0.762, −0.251; P: 0.01, 0.01, respectively) confirmed H6.

The four path coefficients of the two structural equations of AL and NL, namely, IS→PB, ES→PB, IS→PR, and ES→PR, have the opposite signs and different values, indicating that

farmers with different livelihood types have different perceptions of the IS and ES and thus have different perceptions of PR and PB of NGC. H7 is confirmed.

**Results analysis of the WNG for the AL farmers.**

1. Load coefficient analysis of the observed variables

First, ES is mainly affected by its current income derived from growing grain (ES3). From the analysis of the load coefficients, the largest of the three observed variables in ES is ES3 (0.84), while the load coefficients of ES1 (0.59) and ES2 (0.60) are relatively small. This shows that for the AL farmers, their current crop income is the most important factor for evaluating the ES. Second, IS was affected by two main factors: the type of livelihood (IS2) and the living expenses

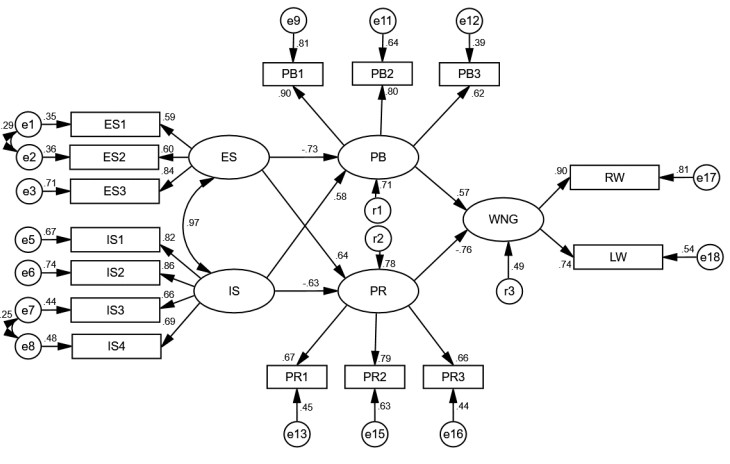

**Fig 4. SEM and standardized path coefficients for AL.**

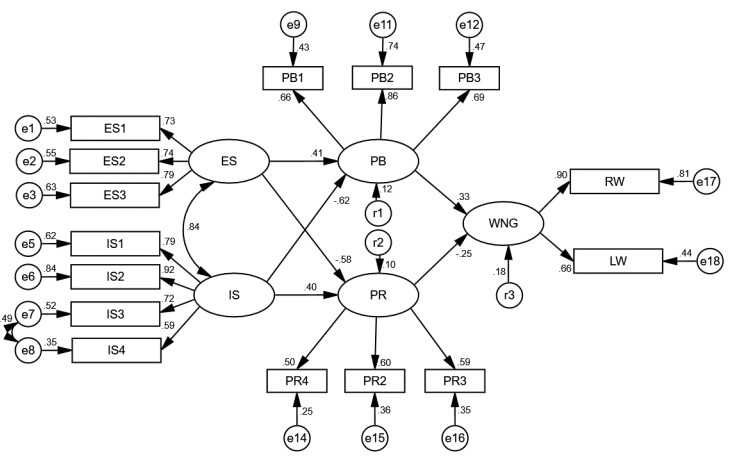

**Fig 5. SEM and standardized path coefficients for NL.**

**Table 5. Path coefficients of SEMs for two types of livelihoods.**

| Path | AL | | | | | NL | | | | |
|---|---|---|---|---|---|---|---|---|---|---|
| | Stan. of nonstandardized estimation results | | | | Stan. path coeff. | Stan. of nonstandardized estimation results | | | | Stan. path coef. |
| | Path coeff. | Stan. error | C.R. | P | | Path coeff. | Stan. error | C.R. | P | |
| ES→PB | −0.642 | 0.187 | −3.441 | *** | −0.729 | 0.322 | 0.137 | 2.354 | ** | 0.414 |
| IS→PB | 0.501 | 0.19 | 2.638 | *** | 0.581 | −0.447 | 0.128 | −3.485 | *** | −0.616 |
| ES→PR | 0.566 | 0.165 | 3.429 | *** | 0.644 | −0.441 | 0.154 | −2.857 | *** | −0.581 |
| IS→PR | −0.541 | 0.202 | −2.676 | *** | −0.625 | 0.309 | 0.132 | 2.337 | ** | 0.405 |
| PR→WNG | −0.57 | 0.11 | −5.161 | *** | −0.762 | −0.199 | 0.075 | −2.657 | *** | −0.251 |
| PB→WNG | 0.436 | 0.121 | 3.606 | *** | 0.573 | 0.212 | 0.046 | 4.635 | *** | 0.332 |
| ES→ES1 | 1 | | | | 0.589 | 1 | | | | 0.73 |
| ES→ES2 | 0.999 | 0.102 | 9.83 | *** | 0.603 | 1.016 | 0.082 | 12.389 | *** | 0.744 |
| ES→ES3 | 1.768 | 0.169 | 10.429 | *** | 0.844 | 1.344 | 0.109 | 12.294 | *** | 0.793 |
| IS→IS1 | 1 | | | | 0.819 | 1 | | | | 0.789 |
| IS→IS2 | 1.08 | 0.066 | 16.321 | *** | 0.863 | 1.33 | 0.078 | 17.102 | *** | 0.918 |
| IS→IS3 | 0.802 | 0.07 | 11.428 | *** | 0.661 | 0.921 | 0.069 | 13.433 | *** | 0.723 |
| IS→IS4 | 0.805 | 0.066 | 12.215 | *** | 0.69 | 0.678 | 0.064 | 10.656 | *** | 0.59 |
| PB→PB1 | 1 | | | | 0.624 | 1 | | | | 0.687 |
| PB→PB2 | 1.406 | 0.135 | 10.456 | *** | 0.797 | 1.327 | 0.124 | 10.718 | *** | 0.861 |
| PB→PB3 | 1.814 | 0.169 | 10.76 | *** | 0.899 | 1.083 | 0.111 | 9.754 | *** | 0.658 |
| PR→PR2 | 1 | | | | 0.794 | 1 | | | | 0.496 |
| PR→PR1 | 0.898 | 0.096 | 9.363 | *** | 0.669 | 1.083 | 0.267 | 4.056 | *** | 0.604 |
| PR→PR3 | 0.425 | 0.077 | 5.488 | *** | 0.664 | — | — | — | — | — |
| PR→PR4 | — | — | — | — | — | 1.159 | 0.237 | 4.898 | *** | 0.594 |
| WNG→RW | 1 | | | | 0.902 | 1 | | | | 0.901 |
| WNG→LW | 0.819 | 0.085 | 9.615 | *** | 0.738 | 0.764 | 0.22 | 3.475 | *** | 0.659 |

(IS1). Among the four observed variables of IS, the load coefficients of IS2 (0.86) and IS1 (0.82) are larger than those of IS3 (0.66) and IS4 (0.69). This phenomenon indicates that the daily living expenses and current livelihood types have greater impacts on the farmers. Third, the economic benefits of growing grain (PB1) and improving the quality of life (PB2) are the most important factors for farmers to evaluate the PB of NGC. Among the four observed variables of PB of NGC, PB4 is not significant, and the load coefficient of PB3 (0.62) is relatively low. The load coefficients of PB1 (0.90) and PB2 (0.80) were relatively high. This shows that farmers believe that NGC has a great impact on increasing grain income and improving the quality of life. Fourth, the risk of technology shortages (PR2) is the most concerning problem when farmers assess the PR of NGC. The load coefficient of PR4 is not significant, the load coefficient of PR1 (0.67) is not high, and the load coefficient of PR3 (0.66) is not high. This shows that farmers are most concerned about a shortage of technology when assessing the perceived risk of NGC.

2. Path analysis of the latent variables

First, the perceived benefit of the farmers promoted the willingness to non-grain conversion of cultivated land, while the perceived risk decreased the willingness to non-grain conversion of cultivated land. According to the standardized path coefficients, the PB (0.57) and PR (−0.76) both affect farmer willingness to NGC of cultivated land. From the perspective of the absolute value of the standardized path coefficient, the influence of PB on the willingness to NGC is greater than that of PB. Second, ES (−0.75) had a negative effect on the PB of NGC for

the farmers, while IS (0.58) had a positive effect on farmers' PB of NGC. The influence of ES is greater than that of IS. Third, ES (0.64) can improve the PR of NGC, while IS (−0.63) can reduce the PR of NGC. At the same time, the impacts of the two methods on the PR of NGC. Fourth, ES has a restraining effect on the WNG of farmers, while IS has an enhancing effect on the WNG. ES can block the PB and enhance the PR of WNG. IS can promote the PB and hinder the PR of WNG.

3. Effect analysis of the latent variables

First, ES negatively affects WNG in an indirect way, and the degree of impact is large (Table 6). The total effect of ES on WNG is −0.90. From the perspective of path, ES affects WNG through two paths: the impact effect of PB is −0.42, and the impact effect of PR is −0.48. Second, IS has an indirect positive impact on WNG, and the degree of impact is large. The total effect of IS on WNG is 0.81. From the path analysis perspective, the impact effect of the IS through the PB is 0.33, and the impact effect through the PR is 0.47. Therefore, IS mainly affects WNG through the PR. Third, the PB directly positively affects WNG, but the degree of influence is moderate. The PB directly affects WNG; there is only one influence path, and the effect is 0.57. Fourth, the PR negatively affected WNG in a direct way, but the degree of influence was high. The PR directly affects WNG, there is only one influence path, and the effect is −0.76.

**Results analysis of the WNG for the NL farmers.**

1. Load coefficient analysis of the observed variables

First, the income from grain farming (ES3) has the greatest impact on ES. According to the load coefficients of the observed variables, the influence of ES3 (0.79) on ES is greater than those of ES1 (0.73) and ES2 (0.74), while the influence of ES4 on ES is not significant. Second, IS was mainly affected by farmer livelihoods (IS2). The load coefficient of IS2 is 0.92, which is greater than those of IS1 (0.79), IS3 (0.72), while the impacts of IS4 (0.59) on IS are relatively low. Third, the quality of life (PB2) is the most important factor affecting PB of NGC. According to the load coefficients, the load coefficient of PB2 (0.86) was greater than those of PB1 (0.66) and PB3 (0.69), while PB4 had no significant effect. This shows that the NL farmers believe that NGC of cultivated land will help improve the life quality of those. Fourth, there are no prominent factors affecting the PR of NGC. The load coefficient of PR2 is only 0.60, while the load coefficients for PR3 (0.59) and PR4 (0.50) are lower. In addition, the load efficient of PR1 is not significant. Such farmers are more worried about the risk of technology shortages after NGC of cultivated land.

2. Path analysis of the latent variables

First, WNG comes from the positive impact of PB and the negative impact of PR of the NGC, and the degrees of impact of both are relatively low. The standardized path coefficient of the PB is 0.33, while that of the PR is −0.25, which indicates that the perceived benefit can promote the willingness to non-grain conversion of cultivated land, while the perceived risk can

**Table 6. The effects of various variables on WNG.**

| | ES | | IS | | PB | | PR | |
|---|---|---|---|---|---|---|---|---|
| | AL | NL | AL | NL | AL | NL | AL | NL |
| Direct effect | — | — | — | — | 0.57 | 0.33 | −0.76 | −0.25 |
| Indirect effect | −0.90 | 0.28 | 0.81 | −0.30 | — | — | — | — |
| Total effect | −0.90 | 0.28 | 0.81 | −0.30 | 0.57 | 0.33 | −0.76 | −0.25 |

block the willingness. Second, ES (0.41) can improve PB of NGC, while IS (−0.62) can reduce PB of NGC. Third, ES (−0.58) can reduce the PR of NGC, while IS (0.40) can increase the PR of NGC. Fourth, ES can enhance PB of NGC but reduce PR of NGC. In contrast, IS can reduce PB and increase PR of NGC.

3. Effect analysis of the latent variables

First, ES positively affect the WNG in an indirect way, and the degree of influence is small. The total effect of ES on the WNG is 0.28. From the perspective of path, ES affects the WNG through two paths: the effect of PB is 0.14, and the effect of PR is also 0.14. Second, IS negatively affects the WNG in an indirect way, and the degree of influence is small. The total effect of IS on WNG is −0.30. From the perspective of path analysis, the influence effect of IS through PB is −0.20, and the influence effect through PR is −0.10. Third, the PB directly positively affects WNG, but the degree of influence is low. The PB directly affects the WNG; there is only one influence path, and the effect is 0.33. Fourth, the PR directly negatively affects the WNG, but the degree of influence is relatively low. The PR directly affects the WNG, and there is only one influence path, with an effect of −0.25.

**Comparative analysis of the two types of farmers.**

1. Load coefficient analysis of the observed variables

First, among the observed variables of ES, ES3 has a greater impact on farmers with agricultural livelihoods. Among the nonagricultural farmers, the influence of ES1 and ES2 is greater than those on the agricultural farmers. At the same time, the influences of ES4 on the farmers with the two livelihood types is not significant. Second, among the observed variables of IS, IS2 has a great impact on the NL farmers, while the impact of IS1 for the AL farmers has increased greatly. IS3 and IS4 had no significant impacts on the two types of farmers. Third, the AL farmers pay attention to the PB of NGC, while the NL farmers pay attention to improving the life quality of those. Improving the social status has no significant effect on the PB of the two types of farmers. Fourth, the AL farmers are more concerned about the risk caused by technology shortages, and the impact of natural disaster risk is not significant. For the NL farmers, the risk of being penalized is not significant, and the impacts of natural disaster risk, technology shortage risk and capital shortage risk are small.

2. Path analysis of the latent variables

First, for the farmers with both types of livelihoods, PB of NGC can increase the WNG, while PR of NGC can block the WNG. PB and PR have greater impacts on RW than LW of WNG. PR has a greater influence on the WNG for the AL farmers. But PB has a greater influence on WNG than does the perceived risk for the NL farmers. This indicates that there are differences in the main influencing factors of WNG with different types of livelihood. Second, there are significant differences in the influencing mechanism of PB of WNG of the different livelihood types of farmers. For the AL farmers, ES reduce PB of NGC, IS increase the PB of NGC, and the former has a greater impact. In contrast, for the NL farmers, ES increased the PB of NGC, IS the reduced the PB of NGC, and the degree of influence was similar. Third, the impact mechanism of the PR of NGC on the farmers with different livelihood types is quite different. ES can improve the PR of NGC for the AL farmers while IS can reduce the PR of NGC for the AL farmers. In contrast, ES reduces the PR of NGC, while IS increases the PR of NGC for the NL farmers.

3. Effect analysis of the latent variables

First, the effect of ES on the WNG varies according to the types of farmer livelihood. ES can greatly reduce the WNG for the AL farmers, while the observed variable can increase

the WNG for the NL farmers, but the degree of impact is low. Second, the effects of survival situations on the WNG vary greatly according to the types of farmer livelihood. IS can enhance the WNG of farmers with AL, but it can block the WNG of farmers with NL. Third, PB of NGC has an enhanced effect on the WNG for the farmers of the two livelihood types, but the degrees of impact are different. The PB of NGC has a great effect on the WNG for the AL farmers, but it has a limited effect on the WNG for the NL farmers. Fourth, the PB of NGC can reduce the WNG of the farmers with two livelihood types, but the degree of impact is different. The PB of NGC can reduce the WNG for the two types of farmers, but the reduction in the WNG is greater for the AL farmers than for the NL farmers.

## Discussion

### Comparison with the existing results

Most of the existing research results explored the NGC of cultivated land from the macro perspectives [13,16]. And this study explored the NGC of cultivated land from the micro perspective, which is from the perspective of farmers. The existing research results do not pay attention to the survival situations of farmers, but these factors are the main influencing factors of farmers' willingness to NGC of cultivated land. In addition, the study incorporated the perceived risk and perceived benefit into the decision-making model of farmers' willingness to NGC of cultivated land, making the research conclusion closer to reality. Therefore, it can provide more practical theoretical support for local government to carry out the policy reform in the field of NGC of cultivated land. In a word, in terms of the research perspective and research content, this study has a certain role in promoting the exploration of NGC of cultivated land

### Limitations and future prospects

This study still has some limitations. First of all, the Chinese government and academia do not have a clear and strict definition of the connotation of NGC of cultivated land. At present, different scholars have the great differences in the definition of NGC of cultivated land, which make the research conclusions less comparable. Secondly, due to the limitation of data collection, the study only analyzed a single cross section data, which may affect our judgment on the influential mechanism of the willingness to NGC of cultivated land.

In the follow-up research, the principle of combining theory and practice to explore the precise connotation of NGC of cultivated land would be focus on. In addition, efforts will made to obtain long-term dynamic data to explore the influential mechanism of survival situations on farmers' willingness to NGC of cultivated land under the different time periods.

## Conclusions and policy implications

### Conclusions

PB of NGC of cultivated land can increase the WNG, while PR of NGC of cultivated land can reduce the WNG. The purposes of NGC of cultivated land for the farmers vary according to their livelihood types. The influential mechanisms of PB about NGC vary according to the farmers' livelihood types. The influential mechanisms of the PR of NGC is different from the livelihood types of farmers. The influences of IS and ES on WNG differ according to the livelihood types of the farmer. These conclusions can provide theoretical support for other major grain-producing provinces in China to improve the policy of controlling the NGC of cultivated land.

## Policy implications

(1) The penalties for NGC of cultivated land should be increases. First, the daily inspection system for NGC of cultivated land should be established, and the phenomenon of NGC of cultivated land can be identified in time. Second, the penalties associated with the laws and regulations for NGC of cultivated land should be increased, which increases the illegal cost of NGC of cultivated land. Third, the local governments should confiscate the part economic benefits of NGC of cultivated land, and the economic benefits of NGC of cultivated land should be reduced.

(2) The policies and supply of funds and technologies for the NGC of cultivated land should be strictly controlled. First, rectification of the relevant policies related to NGC of cultivated land should be implemented immediately from the source to control the phenomenon of the phenomenon. Second, the government or market supply of funds, technology and other fields related to NGC of cultivated land should be strictly controlled, and it is necessary of scientifically assessing for the negative impacts on the protection of cultivated land and food security.

(3) According to the types of farmers'livelihood, different management and control measures should be taken. For the farmers with agricultural livelihoods, the keys to restraining the willingness of non-grain conversion of cultivated land are to increase grain subsidies, reduce living expenses and enhance scientific and technological support for grain production. However, for the farmers with non-agricultural livelihood, restraining the willingness of non-grain conversion of cultivated land mainly lies in enhancing policy control, optimizing livelihood structure and improving quality of life.

(4) The local governments should clarify the attitudes toward NGC of cultivated land. First, it is strictly forbidden for local governments to adopt a tacit attitude towards the NGC of cultivated land. The governments should pay more attention to the protection of cultivated land and national food security rather than increasing local fiscal revenue by the NGC of cultivated land. Second, the management and control of NGC should be included in the assessment scopes of local governments, and the assessment standards should be clarified.

(5) It is necessary to increase grain income through multiple channels. On the one hand, it should reduce the cost of growing grain and reduce the risk of growing grain. On the other hand, governments expand the channels of increasing grain income. Governments should pay attention to the development of agricultural science and technology systems. It is very important to increase investment in agricultural science and technology. More enterprises should be encouraged to participate in agricultural science and technology research and development.

## Supporting information

**S1 Fig. SEM and standardized path coefficients for AL.**
(XLSX)

**S2 Fig. SEM and standardized path coefficients for NL.**
(XLSX)

**S1 Data. Raw data of manuscript.**
(XLSX)

## Author contributions

**Conceptualization:** Hui Fan.

**Data curation:** Jiaxin Liu.

**Formal analysis:** Chaoge Shi.

**Funding acquisition:** Hui Fan.

**Investigation:** Yilin Yang.

**Methodology:** Jiaxin Liu.

**Project administration:** Yilin Yang.

**Software:** Jiaxin Liu.

**Supervision:** Bingqian Sun.

**Validation:** Xiaoke Shao, Bingqian Sun.

**Writing – original draft:** Hui Fan.

**Writing – review & editing:** Xiaoke Shao.

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
