## [Decision Letter · Decision Letter 0]

18 Jul 2024

PONE-D-24-22619The influence of survival conditions on farmer willingness to participate in non-grain conversions of cultivated land based on the SOR modelPLOS ONE

Dear Dr. Fan,

Thank you for submitting your manuscript to PLOS ONE. After careful consideration, we feel that it has merit but does not fully meet PLOS ONE’s publication criteria as it currently stands. Therefore, we invite you to submit a revised version of the manuscript that addresses the points raised during the review process.

 Please see the comments of reviewers below. 

We look forward to receiving your revised manuscript.

Kind regards,

Mingming Li

Academic Editor

PLOS ONE

 [the National Natural Science Foundation of China (Grant number 71904150, 41771438) and General Project of Humanities and Social Sciences Research in Higher Education Institutions in Henan Province（Grant number 2024-ZZJH-039].  

Reviewers' comments:

Reviewer's Responses to Questions

**Comments to the Author**

1. Is the manuscript technically sound, and do the data support the conclusions?

Reviewer #1: Partly

Reviewer #2: Partly

2. Has the statistical analysis been performed appropriately and rigorously? 

Reviewer #1: I Don't Know

Reviewer #2: Yes

3. Have the authors made all data underlying the findings in their manuscript fully available?

Reviewer #1: Yes

Reviewer #2: No

4. Is the manuscript presented in an intelligible fashion and written in standard English?

Reviewer #1: No

Reviewer #2: No

5. Review Comments to the Author

Reviewer #1: Based on SOR model, this paper developed an empirical study to investigate farmer willingness to participate in non-grain conversions of cultivated land in Henan province, China. In general, the topic of this study has some implications for rural land use and farmer livelihood development, but still requires several revisions. Below are some specific comments:

1. The abstract does not provide a concise and clear overview of the results

2. In line 60, the use of "we" is not appropriate for academic writing. Additionally, the overall academic writing style of this manuscript requires improvement.

3. In the second paragraph of the introduction section, the author provides a literature review of the non-grain crop (NGC) of cultivated land from four aspects. However, the influencing factors and driving mechanisms are presented in an overlapping manner, causing confusion for the reader. It is important to note that a literature review is not merely a list of sources but rather a critical evaluation and synthesis of previous works. Therefore, the author should clarify and restructure the presentation of these factors and mechanisms to enhance the logical flow and readability of the introduction.

4. What is the scientific question of your research that is not mentioned in the introduction?

5. The innovation of the article is not clear enough from your literature review.

6. Theoretical analysis and research hypothesis section is not analyzed enough and needs to be further strengthened, the theoretical analysis needs to be supported by certain reviews and theories. It is also not recommended to put “Overview of the SOR model” in the theoretical analysis part.

7. In Section 3.1 on Data Sources, the authors have not provided detailed validation or justification for their conclusion that the data credibility is high.

8. Nowadays, rural households are no longer choosing between the traditional agricultural or non-agricultural livelihoods, but more and more are adopting a combination of agricultural and non-agricultural livelihood strategies. On what basis the author divides these two categories of rural households, the article does not explain in detail, and the division into agricultural and non-agricultural only seems to have some subjective bias.

9. Lines 243-256, from Table 2, how the authors come up with these results, it seems that the results from the table are contrary and that there is an error in the authors' expression

10. The results section lacks concise and clear presentation, and it is recommended that more precise language be used to describe the findings of the study.

11. The discussion section, which requires the author to analyze and compare the main results in depth, is totally missing and discussion is not a reporting of results.

Reviewer #2: The structure of the paper is reasonable, the data processing work is rich, and it has certain innovation. However, the overall article is not smooth, and some of the content description lacks logic. In particular, the expression of the analysis part of the research results is not clear enough, and the scientific problems and main points of the article need to be highlighted. It need a major revision before accepted and can be improved from the following aspects

1.There is a lack of demonstration on the necessity of the influence of individual farmers and family factors on the non-grain conversions of cultivated land. It is suggested to increase in the introduction to reflect the significance of the research.

2.It is suggested to supplement the research status of non-grain conversions of cultivated land in Henan Province.

3.There is a lack of a full description of the abbreviations of various key words in the article.

4.There is a problem with the format of Hypothesis 7 last sentence, please correct it. In addition to this, there are other places in the article also exist the same problem, please check carefully.

5.The technical method part of the article is not clear enough, and it is recommended to add a technical flow chart as a whole.

6.Whether the SOR method is appropriate and reasonable here, it is recommended to supplement the discussion.

7.It is necessary to emphasize the correspondence between policy impact and key conclusions.

6. PLOS authors have the option to publish the peer review history of their article (what does this mean? ). If published, this will include your full peer review and any attached files.

**Do you want your identity to be public for this peer review?** For information about this choice, including consent withdrawal, please see our Privacy Policy .

Reviewer #1: No

Reviewer #2: No

---

## [Author Response · Author response to Decision Letter 1]

3 Oct 2024

Reviewer #1

1. The abstract does not provide a concise and clear overview of the results.

Reply: Thanks for your suggestion. We have revised some contents of the abstract, especially the conclusion part, to improve the generality of the research conclusions. (Page 2, Lines 29-36)

2. In line 60, the use of "we" is not appropriate for academic writing. Additionally, the overall academic writing style of this manuscript requires improvement.

Reply: Thanks for your careful checks. We have revised several similar problems in the paper. (Page 3, Lines 58-59; Page 23 Line 481; Page 23 Lines 484-485)

3. In the second paragraph of the introduction section, the author provides a literature review of the non-grain crop (NGC) of cultivated land from four aspects. However, the influencing factors and driving mechanisms are presented in an overlapping manner, causing confusion for the reader. It is important to note that a literature review is not merely a list of sources but rather a critical evaluation and synthesis of previous works. Therefore, the author should clarify and restructure the presentation of these factors and mechanisms to enhance the logical flow and readability of the introduction.

Reply: Thank you for your nice suggestion. We have combined and adjusted the two parts of influencing factors and driving mechanism appropriately. (Page 3, Lines 66-75)

4. What is the scientific question of your research that is not mentioned in the introduction?

Reply: Thanks for your suggestion. In writing, we neglected this problem. Now, we have added scientific questions in the preface, which improves the readability of the article. (Page 5, Lines 103-105)

5. The innovation of the article is not clear enough from your literature review.

Reply: We sincerely appreciate the valuable comments. We have made several changes in the introduction to optimize the content and highlight the innovation of the article. (Page 5, Lines 97-111)

6. Theoretical analysis and research hypothesis section is not analyzed enough and needs to be further strengthened, the theoretical analysis needs to be supported by certain reviews and theories. It is also not recommended to put “Overview of the SOR model” in the theoretical analysis part.

Reply: Thank you for the suggestion. We have revised this part. The introduction of SOR model is integrated into the specific content of the paper. (Pages 6-7, Lines 123-160)

7. In Section 3.1 on Data Sources, the authors have not provided detailed validation or justification for their conclusion that the data credibility is high.

Reply: Thank you for your nice suggestion. It is supplemented in section 3.1.In order to ensure the accuracy of the questionnaire data, the research group adopted a one-to-one survey when conducting the questionnaire survey, and excluded the questionnaires with missing observational values, inappropriate options and other potential problems. (Page 10, Lines 227-230)

8. Nowadays, rural households are no longer choosing between the traditional agricultural or non-agricultural livelihoods, but more and more are adopting a combination of agricultural and non-agricultural livelihood strategies. On what basis the author divides these two categories of rural households, the article does not explain in detail, and the division into agricultural and non-agricultural only seems to have some subjective bias.

Reply: Thanks for your suggestion. In the previous interviews and investigations, the research group found that family livelihood is the main symbol to distinguish farmers in the research area, and farmers with different family livelihoods have great differences in their willingness to convert cultivated land to non-grain land. Therefore, this study explores the non-grain willingness of cultivated land from the perspective of livelihood, combined with the actual situation of the research area. (Page 5, Lines 105-110)

9. Lines 243-256, from Table 2, how the authors come up with these results, it seems that the results from the table are contrary and that there is an error in the authors' expression.

Reply: Thank you for your nice suggestion. The research group rechecked this part of the manuscript, and there was no problem. The reviewer's question is mainly because our expression may not be appropriate. In understanding this part of the text, Please combine Table 2. Measurement variables and descriptive statistics, especially according to the Observation Variables meaning and value of question items.

For example, the mean value (2.52) of ES1 for AL is small than that (2.72) of ES1 for NL. This shows that ES1 has a greater impact on. The mean value (2.52) of ES1 for al is small than that (2.72) of ES1 for NL. This shows that ES1 has a greater impact on AL farmers. We have revised the statistical analysis results of the perceived benefits and perceived risks of non-grain cultivation.

10. The results section lacks concise and clear presentation, and it is recommended that more precise language be used to describe the findings of the study.

Reply: Thanks for your suggestion. We have modified the result, deleted some unnecessary expressions, and improved the simplicity of the language expression in this part. (Page 24, Lines 491-496)

11. The discussion section, which requires the author to analyze and compare the main results in depth, is totally missing and discussion is not a reporting of results.

Reply: Thank you for the suggestion. We have revised this part. The previous summative content has been deleted to highlight the comparison between the research results of this paper and related results. (Page 23, Lines 481, 484-485)

Reviewer #2:

1. There is a lack of demonstration on the necessity of the influence of individual farmers and family factors on the non-grain conversions of cultivated land. It is suggested to increase in the introduction to reflect the significance of the research.

Reply: Thank you for your nice suggestion. In the introduction, we added the literature in the field of the influence of individual factors and family factors of farmer willingness to convert cultivated land to non-grain land. (Page 5, Lines 97-103)

2. It is suggested to supplement the research status of non-grain conversions of cultivated land in Henan Province.

Reply: Thanks for your suggestion. In the introduction, we added the literature on the current situation of non-grain conversion of cultivated land in Henan Province. (Page 5, Lines 113-115)

3. There is a lack of a full description of the abbreviations of various key words in the article.

Reply: Thank you for the suggestion. We have added various abbreviations to the attachments of the article. (Page 26, Lines 545-551)

4. There is a problem with the format of Hypothesis 7 last sentence, please correct it. In addition to this, there are other places in the article also exist the same problem, please check carefully.

Reply: Thank you for your nice suggestion. We corrected this mistake, and also corrected other similar mistakes in the paper. (Page 9, Lines 205-206)

5. The technical method part of the article is not clear enough, and it is recommended to add a technical flow chart as a whole.

Reply: Thanks for your suggestion. We have added a technical flow chart to make the technical methods section clearer. (Page 9, Line 210, Figure 2)

6. Whether the SOR method is appropriate and reasonable here, it is recommended to supplement the discussion.

Reply: Thank you for the suggestion. We explain the application of SOR in this article. (Pages 6-7, Lines 136-141)

7. It is necessary to emphasize the correspondence between policy impact and key conclusions.

Reply: Thank you for your nice suggestion. We have revised the policy recommendations section to provide a better correspondence between policy impact and conclusions. (Page 25, Lines 512-519, 527-533)

---

## [Decision Letter · Decision Letter 1]

15 Dec 2024

PONE-D-24-22619R1The influence of survival conditions on farmer willingness to participate in non-grain conversions of cultivated land based on the SOR modelPLOS ONE

Dear Dr. Fan,

Thank you for submitting your manuscript to PLOS ONE. After careful consideration, we feel that it has merit but does not fully meet PLOS ONE’s publication criteria as it currently stands. Therefore, we invite you to submit a revised version of the manuscript that addresses the points raised during the review process.

We look forward to receiving your revised manuscript.

Kind regards,

Mingming Li

Academic Editor

PLOS ONE

Journal Requirements:

**Additional Editor Comments:**

Though the other two reviewers give positive comments, the third author has some concerns on your sample selection. Please give more explanation on why the data is representative and has general meaning potentially.

Reviewers' comments:

Reviewer's Responses to Questions

**Comments to the Author**

1. If the authors have adequately addressed your comments raised in a previous round of review and you feel that this manuscript is now acceptable for publication, you may indicate that here to bypass the “Comments to the Author” section, enter your conflict of interest statement in the “Confidential to Editor” section, and submit your "Accept" recommendation.

Reviewer #1: All comments have been addressed

Reviewer #3: All comments have been addressed

Reviewer #4: All comments have been addressed

2. Is the manuscript technically sound, and do the data support the conclusions?

Reviewer #1: Yes

Reviewer #3: Partly

Reviewer #4: Yes

3. Has the statistical analysis been performed appropriately and rigorously? 

Reviewer #1: I Don't Know

Reviewer #3: I Don't Know

Reviewer #4: Yes

4. Have the authors made all data underlying the findings in their manuscript fully available?

Reviewer #1: Yes

Reviewer #3: Yes

Reviewer #4: Yes

5. Is the manuscript presented in an intelligible fashion and written in standard English?

Reviewer #1: Yes

Reviewer #3: Yes

Reviewer #4: No

6. Review Comments to the Author

Reviewer #1: (No Response)

Reviewer #3: The authors have improved their paper adding important parts. Still I have doubts concerning the impact of their research. The case study, even based on 650 questionnaires, could be not essential for the China Policy because involves only one Province. Also for the world this could be a very specific case. Authors tried to explain in their specific way a simple case that could not be useful. Maybe this must be added at the Conclusion chapter. Or specified since the beginning. The perceived benefit is always first, that is why the research must be orientated in other direction.

Reviewer #4: Accept it as it is. The manuscript is appropriate and follows journal regulations and guidelines. This is a very important issue.

7. PLOS authors have the option to publish the peer review history of their article (what does this mean? ). If published, this will include your full peer review and any attached files.

**Do you want your identity to be public for this peer review?** For information about this choice, including consent withdrawal, please see our Privacy Policy .

Reviewer #1: No

Reviewer #3: No

Reviewer #4: **Yes: ** Dr. Ioannis Adamopoulos

---

## [Author Response · Author response to Decision Letter 2]

13 Jan 2025

In response to the reviewers' comments, we have provided the following answers to the relevant questions that require a response

2. Is the manuscript technically sound, and do the data support the conclusions?

Drawing on relevant articles, we observe the corresponding principles in the process of questionnaire design, questionnaire survey, and model construction in order to ensure that the scientific validity of the findings is guaranteed

3. Has the statistical analysis been performed appropriately and rigorously?

In order to make the statistical analysis more scientific and rigorous, we used a questionnaire survey method combining stratified sampling and random sampling. The process of statistical analysis of the data also strictly adhered to the requirements of statistics.

5. Is the manuscript presented in an intelligible fashion and written in standard English?

We have invited scholars with experience of studying in the United States to revise the language of the article during the writing process, and in addition we have asked a specialized AJE agency to polish the language of this article.

6. Reviewer #3: The authors have improved their paper adding important parts. Still I have doubts concerning the impact of their research. The case study, even based on 650 questionnaires, could be not essential for the China Policy because involves only one Province. Also for the world this could be a very specific case. Authors tried to explain in their specific way a simple case that could not be useful. Maybe this must be added at the Conclusion chapter. Or specified since the beginning. The perceived benefit is always first, that is why the research must be orientated in other direction.

Thank you for your suggestions. We understand your concerns about the generalizability and practical impact of the study. Indeed, our case study is limited to a single province, so its implications for policy in China may be limited, and there are some limitations for global replication. Through a detailed examination of Henan Province, a major food-producing region, this study is dedicated to exploring the factors influencing the willingness to NGC of cultivated land and its specificities, with a view to providing theoretical support and lessons for the governance of defragmentation of arable land in other similar regions. We have already pointed this out in the introduction (lines 122-126) and in the conclusion of this paper (lines 501-503). And we state the scope of application of the conclusions of this study, which mainly applies to China's main grain-producing provinces.

---

## [Editor Report · Decision Letter 2]

16 Jan 2025

The influence of survival conditions on farmer willingness to participate in non-grain conversions of cultivated land based on the SOR model

PONE-D-24-22619R2

Dear Dr. Fan,

We’re pleased to inform you that your manuscript has been judged scientifically suitable for publication and will be formally accepted for publication once it meets all outstanding technical requirements.

Kind regards,

Mingming Li

Academic Editor

PLOS ONE
---

## [Editor Report · Acceptance letter]

PONE-D-24-22619R2

PLOS ONE

Dear Dr. Fan,

I'm pleased to inform you that your manuscript has been deemed suitable for publication in PLOS ONE. Congratulations! Your manuscript is now being handed over to our production team.

Kind regards,

on behalf of

Dr. Mingming Li

Academic Editor

PLOS ONE